# Skeletal Muscle Gene Expression Profile in Response to Caloric Restriction and Aging: A Role for SirT1

**DOI:** 10.3390/genes12050691

**Published:** 2021-05-05

**Authors:** Matthew J. Myers, Fathima Shaik, Fahema Shaik, Stephen E. Alway, Junaith S. Mohamed

**Affiliations:** 1Laboratory of Muscle Biology and Sarcopenia, Division of Exercise Physiology, Department of Human Performance, West Virginia University School of Medicine, Morgantown, WV 26506, USA; mjmyers@mix.wvu.edu (M.J.M.); salway@uthsc.edu (S.E.A.); 2Biochemistry and Cellular and Molecular Biology Program, University of Tennessee, Knoxville, TN 37996, USA; fshaik2@vols.utk.edu (F.S.); fshaik3@vols.utk.edu (F.S.); 3Laboratory of Muscle Biology and Sarcopenia, Department of Physical Therapy, College of Health, Professions, University of Tennessee Health Science Center, Memphis, TN 38163, USA; 4Center for Muscle, Metabolism, and Neuropathology, Division of Rehabilitation Sciences, College of Health Professions, University of Tennessee Health Science Center, Memphis, TN 38163, USA; 5Laboratory of Muscle and Nerve, Department of Diagnostic and Health Sciences, College of Health Professions, University of Tennessee Health Science Center, Memphis, TN 38163, USA

**Keywords:** sirtuin1, RNA-sequencing, transcriptome, caloric restriction, SirT1 knockout mice, SirT1 transgenic mice, aging

## Abstract

SirT1 plays a crucial role in the regulation of some of the caloric restriction (CR) responsive biological pathways. Aging suppresses SirT1 gene expression in skeletal muscle, suggesting that aging may affect the role of CR in muscle. To determine the role of SirT1 in the regulation of CR regulated pathways in skeletal muscle, we performed high-throughput RNA sequencing using total RNA isolated from the skeletal muscles of young and aged wild-type (WT), SirT1 knockout (SirT1-KO), and SirT1 overexpression (SirT1-OE) mice fed to 20 wk ad libitum (AL) or 40% CR diet. Our data show that aging repressed the global gene expression profile, which was restored by CR via upregulating transcriptional and translational process-related pathways. CR inhibits pathways linked to the extracellular matrix and cytoskeletal proteins regardless of aging. Mitochondrial function and muscle contraction-related pathways are upregulated in aged SirT1 KO mice following CR. SirT1 OE did not affect whole-body energy expenditure or augment skeletal muscle insulin sensitivity associated pathways, regardless of aging or diet. Overall, our RNA-seq data showed that SirT1 and CR have different functions and activation of SirT1 by its activator or exercise may enhance SirT1 activity that, along with CR, likely have a better functional role in aging muscle.

## 1. Introduction

Sir2 (silent information regulator 2), the mammalian homolog of SirT1 (Sirtuin 1), was originally identified in budding yeast, *Saccharomyces cerevisiae*, as a key regulator of lifespan [1,2]. Interestingly, studies from other lower model organisms such as worms and flies also proved the role of Sir2 homologs in governing lifespan [3]. To date, seven members (SirT1-7) of the evolutionarily conserved homologs to yeast Sir2 have been identified in mammals. All sirtuins are NAD^+^-dependent protein deacetylases, which are involved in numerous pathophysiological roles such as aging, metabolism, cancer, skeletal muscle adaptation, inflammation, and neurodegeneration [4,5,6,7,8,9,10,11,12,13]. The most studied member of the mammalian sirtuin family is SirT1. Although SirT1 was originally identified as one of the deacetylases of histone proteins, SirT1 deacetylates many non-histone proteins. The first-identified non-histone target for SirT1 was p53, which activity is inhibited by SirT1-mediated deacetylation-dependent mechanisms upon DNA damage or oxidative stress, resulting in reduced apoptosis [14,15]. Another important target of SirT1 protein is the regulator of mitochondrial biogenesis peroxisome proliferator-activated receptor-γ co-activator 1α (PGC1α), which activity is increased by reversible acetylation thereby SirT1 promotes mitochondrial biogenesis [9,11,12,16]. One of the physiological stimuli that robustly activates SirT1 is caloric restriction (CR).

CR is a dietary regimen that reduces food intake without incurring malnutrition. While excess dietary intake accounts for the increase in metabolic disorders, CR promotes metabolic fitness, extends lifespan, and disease protection [17,18,19,20,21,22,23,24]. Both SirT1 levels and activity are increased in response to CR followed by extended lifespan in lower organisms such as yeast, worm, and flies [1,2,25,26,27,28]. While SirT1 does not significantly increase lifespan in mice, SirT1 overexpression does improve healthspan [29]. Overall, it is clear that health and life span can be modulated by CR and SirT1 plays a major role in promoting the effect of CR in various model organisms and appears to do the same in humans [30]. The effect of CR varies between various tissues and organs. For example, CR in rodents seems to either decrease or delay the onset of many age-related changes in skeletal muscle [31]. A short-term CR increased muscle stem-cell (satellite cell) numbers and activity, both of which are reduced during aging [13,32,33,34], in both young and old male mice [35].

Skeletal muscle, due to its proportional mass and physical and metabolic functions, is an excellent model to study tissue aging, especially in the context of CR [36]. Despite the role of CR in the regulation of health and life span, it is not clear that how CR controls the global gene expression profile with and without the function of SirT1 in skeletal muscle. Furthermore, the impact of aging on the CR responsive global gene expression profile in skeletal muscle is unknown. Identifying the global gene expression profile and associated biological pathways controlled by CR during aging will allow us to further understand whether CR has a beneficial effect in slowing or reversing sarcopenia, an age-associated loss of muscle mass, quality, and strength. We have previously shown that aging represses both SirT1 gene expression and activity in skeletal muscle [10]. The objective of this study was to investigate whether SirT1 gain-of-function will preserve some of the CR functions in skeletal muscle during aging.

## 2. Materials and Methods

### 2.1. Animals

In this study, we used three different phenotypes of mice. All mice were obtained from Jackson Laboratory (Bar Harbor, ME, USA). Wild-type (WT) C57BL/6J mice (Stock No: 000664). We generated SirT1 knockout mice (KO) mice by crossing SirT1 flox mice (B6;129-Sirt1tm1Ygu/J; Stock No: 008041) with muscle creatine kinase (MCK) promoter-driven Cre mice (B6.FVB(129S4)-Tg(Ckmm-cre)5Khn/J; Stock No: 006475). The resulting KO mice will have exon 4 deleted SirT1 protein in skeletal muscle. For SirT1 overexpression mice (OE), we used B6.Cg-Tg(Sirt1)ASrn/J mice (Stock No: 024510). These mice express a WT mouse SirT1 gene directed by its endogenous promoter/enhancer regions on the BAC transgene. All these mice were maintained per the National Institutes of Health Guide for the Care and Use of Laboratory Animals. The Institutional Animal Care and Use Committee of the West Virginia University (WVU) approved the animal protocols (1603001722; 04/18/2017). The mice were kept in a temperature-controlled room on a 12 h light/dark cycle, with 60% humidity, and food and water ad libitum. We used two age groups of mice, adult (20 weeks) and aged (80 weeks). We used equal number of males and female mice (6–8 mice per group).

### 2.2. Dietary Regime

We used a similar CR diet protocol as published before [22]. Briefly, we used two dietary groups of animals, ad libitum (AL) and CR. To calculate CR, we first measured the quantity of daily AL dietary intake of standard chow diet for one week before starting the CR diet regime. We then reduced the daily food allotment of 60% of that eaten by the AL animals to CR animals for 20 days.

### 2.3. Transcriptome Sequencing and Data Analysis

For RNA-sequencing, we used total RNA isolated from the gastrocnemius (GA) muscles of AL and CR fed animals using Trizol Reagent (Thermo Fisher Scientific, Pittsburgh, PA, USA) followed by the RNeasy Plus Mini Kit (Qiagen, Germantown, MD, USA) as per the manufacturer instructions. We used Bioanalyzer 2100 (Agilent, Califonia, CA, USA) and ND-1000 Spectrophotometer (Thermo Fisher Scientific) to estimate the RNA integrity number (RIN) number and the total RNA that had only the RIN number ≥ 8.0 in the library preparation. Briefly, we utilized ~1 mg of total RNA, representing a specific adipose type, to deplete ribosomal RNA according to the Epicentre Ribo-Zero Gold Kit (Illumina, San Diego, CA, USA). After purification, the poly(A)- or poly(A)+ RNA fractions was fragmented into small pieces 300 bp (±50 bp) using divalent cations under high temperature. Then, we reverse-transcribed the cleaved RNA fragments to create the final cDNA library as per the manufacturer’s protocol for the mRNA-Seq sample preparation kit (Illumina). We performed the paired-end sequencing on an Illumina’s MiSeq at WVU Genome Core Facility.

We used FastQC (http://www.bioinformatics.babraham.ac.uk/projects/fastqc/) to check quality control of raw sequence data and the Trimmomatic flexible read trimming tool [37] to remove the adapter and other Illumina-specific sequences as well low-quality bases, and undetermined bases from the raw read. We used HISAT2 [38] to map the trimmed and clean reads against the *Mus musculus* genome (mm10). Using featureCounts [39], we measured gene expressions of all the mapped reads of each sample. Differential gene expressions (DGE) between the two groups were performed by limma-voom [40,41,42,43]. StringTie [44] was used to perform expression levels for mRNAs and lncRNAs by calculating fragments per kilobase of transcript per million mapped reads (FPKM). The raw RNA sequencing data can be found at the NCBI Sequence Read Archive database under the BioProject accession number PRJNA702674 (SAMN14838464–SAMN14838469) and SRA accession number (SRR13764286–SRR13764377).

### 2.4. GO and KEGG Analyses

We used g:Profiler [45], a free public web server for characterizing and manipulating gene lists resulting from mining high-throughput genomic data, to find biological pathways, cellular components, molecular functions, subcellular localization, and shared TF-binding sites (TFBS) for the differentially expressed genes. The data used in g:Profiler are derived from the Gene Ontology [46], KEGG [47], Reactome [48] and TRANSFAC [49] databases.

### 2.5. Custom Functional Analysis

We performed an exhaustive search on PubMed for the chosen differentially expressed genes with key terms “skeletal muscle”, “caloric restriction”, “metabolism”, and/or “mitochondria”. The initial few pages of relevant search results were carefully examined, and the pertinent literature reviewed to elucidate the known and potential functional gene roles related specifically to skeletal muscle homeostasis.

## 3. Results

### 3.1. Summary of Skeletal Muscle Gene Expression Profile in 20 and 80 wk Old WT, SirT1-KO, and SirT1-OE Mice

The workflow of library preparation and RNA sequencing as well as bioinformatic analyses are shown in Figure 1. To determine the role of CR and SirT1 in the regulation of age-associated global gene expression profile and related biological pathways, we used total RNAs from the GA muscles of 20- and 80-week-old WT, SirT1-KO, and SirT1-OE mice following 20 days of AL or CR feeding regimen. We used the RNA samples with the RIN number ≥ 8.0 for library preparation. After depletion of ribosomal RNA, followed by fragmentation with divalent cation in elevated temperature, we prepared 92 sequencing libraries for 12 groups using Illumina’s TruSeq-stranded-total-RNA-sample preparation protocol. Each group contained six to eight libraries (n = 6–8). Table 1 shows the database and bioinformatics software used in this study. Briefly, we obtained a total of ~17 to ~18 million raw reads from 92 samples. After removing adaptors, contamination, and low-quality reads, we selected the best four quality paired-end reads from each group (n = 4), which represented a total of 48 reads from 12 groups (4 samples/group). After mapping these valid reads against the *Mus musculus* genome, we obtained a range of 5.7 to 10.3 million reads through the samples.

### 3.2. Aging Augments Muscle Contraction and Cellular Transcription and Translation Pathways and Represses Ubiquitin and Energy Homeostasis Related Pathways in the Skeletal Muscle of WT Mice

To determine the effect of aging in the regulation of skeletal muscle gene expression profile, we performed DGE analysis in young and aged WT mice. DGE analysis identified 11,487 differentially expressed genes in the skeletal muscle of WT mice in response to aging (Appendix A). The log2-fold changes and statistical significance of each gene calculated by performing a DGE analysis are shown as a volcano plot (Appendix A). DGE analysis determined a total of 74 significantly (*p* ≤ 0.05) expressed genes in which 39 genes were upregulated and 35 genes were downregulated (Table 2 and Figure 2A). However, only 10 genes were differentially expressed ≥2-fold, which all were upregulated, and no genes were downregulated ≥2-fold in WT mice in response to aging (Table 3). These results suggest that aging robustly diminished the global gene expression profile in the skeletal muscle. To determine the biological significance of the differentially expressed genes, we performed GO analysis with one or more GO terms. All GO terms were divided into three categories: “biological process (BP)”, “cellular component (CC)”, and “molecular function (MF)”. A total of 624 pathways were significantly regulated at *p* ≤ 0.05 with 364 and 260 pathways were up and downregulated, respectively (Appendix A and Figure 3A). The top upregulated pathways were associated with muscle contraction such as “muscle filament sliding”, “actin-myosin filament sliding”, “muscle contraction”, “striated muscle contraction”, “skeletal muscle contraction”, “actin binding”, “microfilament motor activity”, “myofibril assembly”, “cytoskeleton”, “actin filament binding”, “regulation of muscle contraction”, “actin cytoskeleton”, and “skeletal muscle thin filament assembly” (Appendix A). Some of the top upregulated pathways were associated with the transcription and translational process (“SRP-dependent co-translational protein targeting to membrane”, “nuclear-transcribed mRNA catabolic process”, “nonsense-mediated decay”, “protein targeting to ER”, “ribosome”, “transcription regulatory region DNA binding”, “rRNA metabolic process”, “polysomal ribosome”, “rRNA processing”, “transcription corepressor activity”, and “WW domain binding”). The top downregulated pathways were associated with ubiquitin and energy homeostasis such as “ubiquitin-protein transferase activity”, “protein ubiquitination”, “Cul3-RING ubiquitin ligase complex”, “cellular glucose homeostasis”, “regulation of energy homeostasis”, and “positive regulation of ATP metabolic process” (Appendix A).

### 3.3. Aging Promotes Cellular Transcriptions and Represses Mitochondrial Functions in the Skeletal Muscle of SirT1 KO Mice

To identify the role of SirT1 in the regulation of the skeletal muscle gene expression profile during aging, we performed DGE analysis in young and aged SirT1 KO mice. DGE analysis identified 11,774 differentially expressed genes in the skeletal muscle of SirT1 KO mice in response to aging (Appendix A). Appendix A shows the volcano plot of the log2-fold changes and statistical significance of each gene calculated by performing a DGE analysis in SirT1 KO mice. DGE analysis identified 1146 genes (145 up and 1001 down) that were significantly expressed at *p* ≤ 0.05 (Table 2 and Figure 2A). There were 23 (7 up and 16 down) genes differentially expressed at ≥2-fold (Table 3). These data suggest that SirT1 loss-of-function considerably changed the global skeletal muscle gene expression profile, particularly many of the genes were downregulated during aging. To determine the biological significance of the differentially expressed genes, we performed GO analysis, which identified 703 significantly regulated (*p* ≤ 0.05) pathways, 508 were upregulated and 195 were downregulated pathways (Appendix A and Figure 3A). The top pathways upregulated in response to SirT1 loss-of-function were associated with cellular transcriptions such as “regulation of transcription from RNA polymerase II promoter”, “transcription regulatory region sequence-specific DNA binding”, “nuclear-transcribed mRNA catabolic process”, “RNA polymerase II regulatory region DNA binding”, “positive regulation of transcription, DNA-templated”, “positive regulation of transcription from RNA polymerase II promoter”, “transcription coactivator activity”, “nuclear-transcribed mRNA catabolic process”, “nonsense-mediated decay”, “transcription regulatory region DNA binding”, “negative regulation of transcription”, and “DNA-templated, negative regulation of transcription from RNA polymerase II promoter” (Appendix A). Though not many pathways were downregulated, some of these pathways are associated with mitochondria such as “basement membrane organization”, “mitochondrion”, “respiratory electron transport chain”, and “mitochondrial inner membrane” (Appendix A).

### 3.4. GTPase Signaling, DNA Binding Activity, and Ion Channel Functions Are Activated Whereas Apoptotic Signaling, and Cellular Metabolism Are Suppressed in SirT1 OE Mice in Aging

We next performed DGE analysis in young and aged SirT1 OE mice to identify whether SirT1 gain-of-function will improve the age-associated deleterious effect by altering skeletal muscle gene expression profile. DGE analysis showed 11,923 differentially expressed genes in the skeletal muscle of SirT1 OE mice in response to aging (Appendix A). Appendix A displays the volcano plot of the log2-fold changes and statistical significance of each gene calculated by performing a DGE analysis in these mice. Among those differentially expressed genes, 284 (61 up and 223 down) genes were significantly expressed at *p* ≤ 0.05 (Table 2 and Figure 2A). The number of genes differentially expressed at ≥2-fold was 3, which were upregulated, and no genes were downregulated at ≥2-fold (Table 3). These data suggest that SirT1 gain-of-function significantly downregulated many genes in response to aging. GO analysis found that 771 pathways were significantly regulated at *p* ≤ 0.05 with 275 and 496 pathways were up and downregulated, respectively (Appendix A and Figure 3A). The leading upregulated pathways were associated with GTPase signaling (“Rab GTPase binding” and “GTPase activator activity”), DNA binding activity (“transcription regulatory region sequence-specific DNA binding”, “core promoter binding”, “core promoter sequence-specific DNA binding”, and “transcription regulatory region DNA binding”) and ion channels (“voltage-gated cation channel activity”, “voltage-gated potassium channel activity”, and “delayed rectifier potassium channel activity”). Secretary pathways like “negative regulation of insulin secretion”, and “negative regulation of protein secretion” also upregulated in SirT1 OE mice during aging (Appendix A). The leading downregulated pathways in response to aging were associated with apoptotic signaling, including “regulation of the apoptotic process”, “positive regulation of extrinsic apoptotic signaling pathway”, “negative regulation of the apoptotic process”, “positive regulation of the apoptotic process”, “positive regulation of extrinsic apoptotic signaling pathway via death domain receptors”, “regulation of extrinsic apoptotic signaling pathway via death domain receptors”, and “positive regulation of apoptotic signaling pathway”. The pathways associated with cellular metabolism such as “positive regulation of the cellular metabolic process”, “glutathione metabolic process”, “lipoprotein particle binding”, “lipoprotein particle receptor activity”, and “positive regulation of cellular biosynthetic process” were also downregulated in SirT1 OE mice (Appendix A).

### 3.5. SirT1 Loss-of-Function Increases Mitochondrial Functions, Metabolism and Ubiquitin Proteasomal Activity, and Decreases Muscle Function and Insulin Signaling in the Skeletal Muscle of Young Mice

To establish the role of SirT1 in the regulation of skeletal muscle gene expression profile, we performed DGE analysis in young WT and SirT1 KO mice. DGE analysis found 11,505 genes following SirT1 loss-of-function in 20-week-old mice. (Appendix A). Appendix A displays the volcano plot of the log2-fold changes and statistical significance of each gene calculated by performing a DGE analysis. DGE analysis determined 566 (80 up and 486 down) significantly (*p* ≤ 0.05) regulated genes in SirT1 KO mice (Table 2 and Figure 2B). Among those significantly regulated genes, only 51 genes were highly differentially expressed (12 up and 39 down) at ≥2-fold (Table 4). These data imply that a significant number of genes were downregulated in skeletal muscle following SirT1 KO. The GO annotation analysis provided 917 differentially regulated pathways (409 up and 508 down) at *p* ≤ 0.05 (Appendix A and Figure 3B). The top upregulated pathways in SirT1 KO mice were associated with mitochondrial functions such as “mitochondrial ATP synthesis coupled electron transport”, “mitochondrial respiratory chain complex I”, “respiratory electron transport chain”, “NADH dehydrogenase (quinone and ubiquinone) activity”, “mitochondrial electron transport, NADH to ubiquinone”, “mitochondrial inner membrane”, “mitochondrial respiratory chain complex assembly”, “mitochondrial respiratory chain complex I biogenesis”, “NADH dehydrogenase complex assembly”, and “mitochondrial respiratory chain complex I assembly” (Appendix A). Some of the top pathways were linked to metabolism such as “regulation of the cellular amino acid metabolic process”, “regulation of cellular ketone metabolic process regulation of cellular ketone metabolic process”, “response to ketone”, and “ubiquitin proteasomal pathway such as regulation of ubiquitin-protein ligase activity”. The top downregulated pathways in the SirT1 KO mice were associated with muscle functions such as “muscle contraction”, “muscle organ development”, “focal adhesion”, and “elastic fiber assembly”. Some of the downregulated pathways were associated with insulin pathways such as “cellular response to insulin stimulus”, “cellular glucose homeostasis”, “3′,5′-cyclic-AMP phosphodiesterase activity”, and “protein phosphorylation” (Appendix A).

### 3.6. SirT1 Gain-of-Function Upregulates Signal Transduction and Downregulates Lipid Metabolism in the Skeletal Muscle of Young Mice

To find the role of SirT1 gain-of-function in the regulation of skeletal muscle gene expression profile, we performed DGE analysis in young WT and SirT1 OE mice. DGE analysis recognized 11,254 genes SirT1 OE mice (Appendix A). The log2-fold changes and statistical significance of each gene calculated by DGE analysis are shown in Appendix A. For significantly (*p* ≤ 0.05) up and down-regulated genes, DGE analysis found 48 genes where 14 genes were upregulated, and 34 genes were downregulated (Table 2 and Figure 2B). Surprisingly, no genes were differentially regulated at ≥2-fold. These results indicate that increased SirT1 activity in skeletal muscle altered a lesser number of genes, which are less than 2-fold. The GO annotation analysis specified 915 differentially regulated pathways (480 up and 435 down) at *p* ≤ 0.05 (Appendix A and Figure 3B). Most of the top upregulated pathways were related to signal transduction, including “protein kinase binding”, “negative regulation of intracellular signal transduction”, “regulation of I-kappaB kinase/NF-kappaB signaling”, “positive regulation of intracellular signal transduction”, “regulation of small GTPase mediated signal transduction”, “transmembrane receptor protein serine/threonine kinase signaling pathway”, and “regulation of extrinsic apoptotic signaling pathway” (Appendix A). The top downregulated pathways were associated with lipid metabolisms such as “lipid droplet”, “lipid homeostasis”, “phosphatidylinositol phosphate phosphatase activity”, “lipid particle organization”, “negative regulation of lipid storage”, “positive regulation of cholesterol transport”, and “phosphatidylinositol-4,5-bisphosphate phosphatase activity”. Some of the downregulated pathways were linked to glucose homeostasis, including “cellular response to nutrient levels”, “cellular glucose homeostasis”, “cellular response to starvation”, “response to insulin”, and “cellular response to insulin stimulus” (Appendix A).

### 3.7. Most of the Top Pathways Altered by CR Are Associated with Translational and Muscle Contraction Process, Which Are Suppressed in the Skeletal Muscle of Young WT Mice

Next, we determined the role of CR in the regulation of skeletal muscle gene expression profile. To do this, we first conducted DGE analysis in young WT mice fed to either an AL or a CR diet. DGE analysis found 11,250 differentially expressed genes in response to CR (Appendix A). The log2-fold changes and statistical significance of each gene calculated by DGE analysis in these groups of mice are shown in Appendix A. There were 802 significantly expressed genes at *p* ≤ 0.05, including 125 upregulated and 677 downregulated genes (Table 2 and Figure 2C). In particular, there were 6 and 41 up and downregulated genes at ≥2.0 fold (*p* ≤ 0.05), respectively (total 47 genes) (Table 5 and Table 6). These data suggest that CR significantly altered a considerable number of genes, which were mostly downregulated. CR altered 917 pathways (up 473 and down 444) in young WT mice (Appendix A and Figure 3C). The top downregulated pathways altered by CR were associated with ribosomal biogenesis and translation such as “protein targeting to ER”, “SRP-dependent co-translational protein targeting to membrane”, “cytosolic ribosome”, “peptide biosynthetic process”, “translation”, “co-translational protein targeting to membrane”, “rRNA processing”, “cytosolic small ribosomal subunit”, “small ribosomal subunit”, “rRNA metabolic process”, “ribosome”, “cellular macromolecule biosynthetic process”, “ribosome biogenesis”, “cytoplasmic translation”, “ncRNA processing”, “cytosolic large ribosomal subunit”, “large ribosomal subunit”, “ribosome assembly”, “ribosomal small subunit biogenesis”, and “polysome” (Appendix A). Some of the downregulated pathways were linked to muscle contraction such as “muscle contraction”, “actin-myosin filament sliding”, “striated muscle contraction”, “focal adhesion”, “regulation of muscle contraction”, “myofibril assembly”, and “cardiac muscle contraction”. The only upregulated pathways appearing in the top 50 altered pathways were “transcription corepressor activity”, “phosphatidylinositol binding”, “lipoprotein particle receptor activity”, and “regulation of cytoskeleton organization” (Appendix A).

### 3.8. CR Promotes Muscle Contraction and Inhibits Mitochondrial Function Related Pathways in the Skeletal Muscle of Young SirT1 KO Mice

Next, we sought to determine the skeletal muscle-specific gene expression profile and associated pathways regulated by CR in the absence of SirT1 activity in young mice. For this, we compared DGE analysis between young WT and SirT1 KO mice fed with either an AL or a CR diet. DGE analysis found 11,553 genes in response to CR (Appendix A). The log2-fold changes and statistical significance of each gene calculated by DGE analysis in these groups of mice are shown in Appendix A. Among those genes, there are 1296 significantly regulated genes at *p* ≤ 0.05, 229 genes were upregulated, and 1067 genes were downregulated (Table 2 and Figure 2C). Notably, there are 32 highly differentially expressed genes (≥2.0 fold at *p* ≤ 0.05) (6 up and 26 down) (Table 5 and Table 6). These data indicate that CR significantly altered a substantial number of genes and a greater number of genes were downregulated. CR altered 999 pathways (684 up and 315 down) in young SirT1 KO mice (Appendix A and Figure 3C). The top upregulated pathways were connected to muscle contraction such as “actin cytoskeleton”, “α-actinin binding, actin-binding”, “actomyosin structure organization”, “sarcomere organization”, “myofibril assembly”, “actin filament binding”, “muscle filament sliding”, “actin-myosin filament sliding”, “striated muscle myosin thick filament assembly”, “actinin binding”, and “striated muscle contraction” (Appendix A). Most of the top downregulated pathways were related to mitochondrial function. These were “mitochondrial ATP synthesis coupled electron transport”, “respiratory electron transport chain”, “mitochondrial inner membrane”, “NADH dehydrogenase (quinone) activity”, “NADH dehydrogenase (ubiquinone) activity”, “mitochondrial electron transport”, “NADH to ubiquinone”, “mitochondrial respiratory chain complex I”, “NADH dehydrogenase complex assembly”, “mitochondrial respiratory chain complex I biogenesis”, “mitochondrial respiratory chain complex I assembly”, “mitochondrial respiratory chain complex assembly”, and “mitochondrial electron transport, cytochrome c to oxygen” (Appendix A).

### 3.9. CR Promotes Gene Expression Events and Inhibits ECM and Cytoskeleton in the Skeletal Muscle of Young SirT1 OE Mice

Next, we compared DGE analysis between young WT and SirT1 OE mice fed with either an AL or a CR diet to determine the role of SirT1 gain-of-function in the regulation of CR responsive genes and related pathways. DGE analysis found 11,207 differentially expressed genes in response to CR (Appendix A). The log2-fold changes and statistical significance of each gene calculated by DGE analysis in these groups of mice are shown in Appendix A. Among those genes, there are 83 and 911 up and down-regulated genes at *p* ≤ 0.05 (total 994 genes), respectively (Table 2 and Figure 2C). Notably, there were 79 genes that were highly expressed (≥2.0 fold at *p* ≤ 0.05) in these mice (2 up and 77 down) (Table 5 and Table 6). These data indicate that CR significantly altered a large number of genes, which were mostly downregulated. CR altered 995 pathways (up 473 and down 444) in young SirT1 OE mice (Appendix A and Figure 3C). The top upregulated pathways were involving in gene expression, especially translational events such as “RNA binding”, “nuclear-transcribed mRNA catabolic process, nonsense-mediated decay”, “SRP-dependent co-translational protein targeting to membrane”, “translation factor activity, RNA binding”, “co-translational protein targeting to membrane”, “protein targeting to ER”, “translation initiation factor activity”, “regulation of translational initiation”, “ribosome biogenesis”, “cytosolic ribosome”, “rRNA processing”, “regulation of translation”, “cytosolic large ribosomal subunit”, “ribonucleoprotein complex assembly”, “ribosome”, “gene expression”, and “ribonucleoprotein complex assembly” (Appendix A). The top downregulated pathways were participating in ECM and cytoskeleton such as “extracellular matrix organization”, “collagen fibril organization”, “protein complex subunit organization”, “focal adhesion”, and “extracellular matrix disassembly”. Some of the downregulated pathways were associated with “endoplasmic reticulum lumen”, “vacuolar lumen”, “lysosomal lumen”, “positive regulation of cell migration”, “positive regulation of cell motility”, and “lysosome” (Appendix A).

### 3.10. CR Positively Regulates Transcription and Translation and Negatively Regulates ECM and Cytoskeleton in the Skeletal Muscle of Aged WT Mice

To define the effect of CR in the aging skeletal muscle gene expression profile, we performed DGE analysis in aged WT mice fed 20 wk an AL or a CR diet. DGE analysis identified 11,487 genes in response to CR (Appendix A). The log2-fold changes and statistical significance of each gene calculated by DGE analysis are shown in Appendix A. DGE analysis identified 92 up and 247 significantly (*p* ≤ 0.05) downregulated genes (total 339 genes) (Table 2 and Figure 2D). However, no genes were differentially expressed ≥2-fold in WT mice. These data suggest that aging hindered CR responsive global gene expression profile in skeletal muscle. CR altered 735 pathways (320 up and 415 down) in aged WT mice (Appendix A and Figure 3D). Most of the top upregulated pathways were associated with transcription and translation such as “SRP-dependent co-translational protein targeting to membrane”, “co-translational protein targeting to membrane”, “protein targeting to ER”, “viral transcription”, “rRNA processing”, “viral gene expression”, “cytosolic ribosome”, “nuclear-transcribed mRNA catabolic process, nonsense-mediated decay”, “ribosome biogenesis”, “rRNA metabolic process”, “cytosolic part”, “ncRNA processing”, “nuclear-transcribed mRNA catabolic process”, “RNA binding”, “peptide biosynthetic process”, “large ribosomal subunit”, “cytoplasmic translation, cytosolic large ribosomal subunit”, “gene expression”, “translation”, “viral process”, “ribosome”, “cytosolic small ribosomal subunit”, “small ribosomal subunit”, “regulation of RNA splicing”, “mRNA processing, and mRNA splicing”, and “via spliceosome” (Appendix A). Most of the top downregulated pathways were connected to ECM and cytoskeleton (“extracellular matrix organization”, “focal adhesion”, “collagen fibril organization”, “integrin binding”, “regulation of cell migration”, “actin cytoskeleton”, “negative regulation of cytoskeleton organization”, and “actin filament binding”; Appendix A).

### 3.11. CR Promotes Mitochondrial β-Oxidation, Fatty Acid Metabolism, and Muscle Contraction and Blocks Cellular Gene Expression in the Skeletal Muscle of Aged SirT1 KO Mice

Next, we sought to determine the effect of CR in skeletal muscle gene expression profile, particularly in the absence of SirT1 during aging. We performed DGE analysis in aged WT and SirT1 KO mice, fed for 20 weeks with an AL or a CR diet. DGE analysis found 11,556 genes in response to CR (Appendix A). The log2-fold changes and statistical significance of each gene calculated by DGE analysis are shown in Appendix A. There are 133 and 332 up and down-regulated genes at *p* ≤ 0.05, respectively (total 465 genes) (Table 2 and Figure 2D). Surprisingly, DGE analysis did not find the genes that were e differentially expressed ≥2.0 fold at *p* ≤ 0.05, except two upregulated genes (Table 7). These data suggest that aging severely affected CR responsive global genes expression profile in skeletal muscle. Moreover, CR changed 869 pathways, including 508 upregulated and 195 downregulated pathways in 80-week-old SirT1 KO mice (Appendix A and Figure 3D). The leading upregulated pathways were associated with mitochondria, (“mitochondrial matrix”, and “mitochondrion”) fatty acid metabolism (“fatty acid β-oxidation”, “fatty acid catabolic process”, “long-chain fatty acid transport”, “regulation of fatty acid oxidation”, “fatty acid oxidation”, and “fatty acid β-oxidation using acyl-CoA dehydrogenase”.) and muscle contraction (“muscle filament sliding”, “actin-myosin filament sliding”, “striated muscle contraction”, “cardiac muscle contraction”, “regulation of cardiac muscle cell contraction”, “contractile fiber”, and “regulation of cardiac conduction”) (Appendix A). The top downregulated pathways were linked to gene transcription and translation such as “regulation of transcription from RNA polymerase II promoter”, “RNA polymerase II regulatory region sequence-specific DNA binding”, “negative and positive regulation of transcription from RNA polymerase II promoter”, “cytosolic ribosome”, “SRP-dependent co-translational protein targeting to membrane”, “transcription corepressor activity”, “ribosome biogenesis”, “protein targeting to ER”, “cytoplasmic translation”, “cytosolic large ribosomal subunit”, “RNA polymerase II core promoter proximal region sequence-specific DNA binding”, “E-box binding”, and “regulation of transcription from RNA polymerase II promoter” (Appendix A).

### 3.12. SirT1 OE Promotes Ubiquitination Signaling and Downregulated Muscle Structure and Contraction Related Proteins Following CR in the Skeletal Muscle of Aged Mice

Next, we determined the effect of CR in gene expression profile following SirT1 gain-of-function in aged muscle. We performed DGE analysis in aged WT and SirT1 OE mice, fed with an AL or a CR diet for 20 weeks. DGE analysis uncovered 11,660 differentially expressed genes in response to CR (Appendix A). Appendix A shows the log2-fold changes and statistical significance of each gene calculated by DGE analysis. CR significantly (*p* ≤ 0.05) regulated 1465 genes, of which 174 genes were upregulated, and 1291 genes were downregulated (Table 2 and Figure 2D). There are 200 highly differentially expressed genes (≥2.0 fold at *p* ≤ 0.05) in response to CR, with 10 upregulated and 190 downregulated (Table 7). These data indicate that CR altered a greater number of genes that were downregulated in aged SirT1 OE mice. Moreover, CR altered 972 pathways, including 411 and 561 up and downregulated pathways, respectively (Appendix A and Figure 3D). The top upregulated biological pathways were related to the ubiquitin-proteasomal pathway. These are “protein ubiquitination”, “ubiquitin-protein transferase activity”, “protein polyubiquitination”, “post-translational protein modification”, “lysosomal transport”, “ubiquitin-dependent protein catabolic process”, “modification-dependent protein catabolic process”, “regulation of cellular component organization”, “endosome to lysosome transport”, “endosome organization”, “ubiquitin-protein ligase activity”, and “cellular protein modification process” (Appendix A). The top downregulated biological pathways were associated with muscle contraction such as “focal adhesion”, “muscle filament sliding”, “actin-myosin filament sliding”, “extracellular matrix organization”, “integrin binding”, “collagen fibril organization”, “muscle contraction”, “actomyosin structure organization”, “myofibril assembly”, “actomyosin”, “actin cytoskeleton”, “actin binding”, “actin monomer binding”, “collagen binding”, “contractile actin filament bundle”, “calcium ion binding”, “regulation of muscle contraction”, “contractile fiber”, “myofibril”, “actin-based cell projection”, “microfilament motor activity”, and “skeletal muscle thin filament assembly” (Appendix A).

## 4. Discussion

CR promotes metabolism, increases lifespan, and prevents or reduces the prevalence of diseases and disorders [17,18,19,20,21,22,23,24]. This beneficial effect of CR is exerted through, in part, SirT1 deacetylase activity [2,27,28]. However, the molecular mechanisms driving these phenomena remain largely unknown. Moreover, we have previously shown that aging suppresses SirT1 gene expression in skeletal muscle [10], suggesting that the loss-of-SirT1 function may reduce the effect of CR in aging muscle. To investigate whether SirT1 gain-of-function will retain the effect of CR in skeletal muscle during aging. To test our hypothesis, we performed whole muscle transcriptome sequencing to identify muscle-specific transcript alterations during aging in a mouse model of CR. We also used SirT1 KO and OE mice to study the role of SirT1 in the regulation of CR responsive genes and associated pathways.

We first compared the skeletal muscle RNA-seq data of young and aged WT, SirT1 KO, and SirT1 OE mice fed to AL diet to determine the effect of aging on the skeletal muscle gene expression profile and the role of SirT1 in this process. Our data show that aging suppresses global gene expression profile as evidenced by a reduction in the total number of differentially expressed genes (74 genes) in the skeletal muscle of WT mice. In contrast, aging robustly altered a vast number of genes (1146) in the skeletal muscle of SirT1 KO mice. However, SirT1 OE mice displayed a very smaller number of genes (284 genes) that are changed in response to the AL diet. Most of the altered genes in SirT1 KO and OE mice are downregulated. Similarly, young SirT1 KO mice showed more altered genes (566 genes) compared to young SirT1 OE mice (48 genes). These data indicate that SirT1 loss-of-function in skeletal muscle severely altered the global gene expression profile, especially in aging that was revered by SirT1 gain-of-function. Studies from our lab and others have shown that aging represses SirT1 gene expression in skeletal muscle and SirT1 activation in aging muscle will improve muscle function [10,11,50,51,52,53]. SirT1 gain-of-function therefore may favor CR to alter global gene expression profile, especially in skeletal muscle. These findings are in agreement with the previous study showed that only 0.9% of genes displayed a greater than two-fold change in expression levels as a function of age [54]. This could be as a result of the fact that the aging process is unlikely to be due to large, widespread alterations in gene expression [54,55]. In WT mice, aging augments the expression of genes associated with muscle contraction and cellular transcriptional and translational events while suppressing ubiquitination and cellular energy homeostasis, including glucose homeostasis and ATP metabolic process. The repressed glucose homeostasis could be due to the reduced PI3K-Akt and AMPK signaling pathways. However, aging represses mitochondrial functions in the skeletal muscle of SirT1 KO mice but not in young SirT1 KO mice. Studies from our lab and others have shown that SirT1 expression and its deacetylase activity in skeletal muscle are dramatically declined in aging [10,56]. In the present study, we did not find either increased or decreased expression of genes related to mitochondrial biogenesis in the skeletal muscle of SirT1 OE mice. This implies that SirT1 may be necessary for regulating mitochondrial function, especially in aging.

SirT1 function is indispensable for CR-induced insulin signaling, which was completely abrogated in mice lacking SirT1 deacetylase activity [57], signifying a role for SirT1 in glucose intake/metabolism. SirT1 KO mice showed a significant reduction in exercise capacity in treadmill and membrane fragility [58] pointing that SirT1 is necessary for muscle activity. These data provide the possibility that the reduced exercise capacity in SirT1 KO mice [58] could be due to the loss of insulin signaling and muscle activity as evidenced by the current RNA-seq data demonstrating decreased muscle function and insulin signaling as well as mitochondrial function in SirT1 KO mice. Another possibility for the loss of muscle function in SirT1 KO mice may be due to the lower density of myonuclei as a consequence of a reduced number of satellite cells (muscle stem cells) [59]. KEGG pathway analysis in SirT1 KO in the present study also showed downregulation of AMPK and PI3K-AKT signaling pathways, which are often activated during insulin signaling, particularly in response to CR [57]. In agreement with this, a previous study has demonstrated that resveratrol and AICAR could not activate AMPK in SirT1 ablated muscle cells and hepatocytes [60]. Although SirT1 has been described many times as a key regulator of mitochondrial biogenesis through the deacetylation of PGC-1α [61], surprisingly, our RNA-seq analysis demonstrated an increase in the expression of genes associated with mitochondrial functions and metabolism in the skeletal muscles of young SirT1 KO mice, but not in the aged mice. In line with this, a study showed that aerobic exercise increased PGC-1α-mediated muscle mitochondrial biogenesis in SirT1 KO mice [62], suggesting that SirT1 may not be necessary for exercise-induced mitochondrial function. This controversy could be due to the likelihood of compensatory deacetylation of PGC-1α by other family members of Sirtuins such as SirT2 [63]. Shockingly, elevated SirT1 protein may impair mitochondrial biogenesis in rodent skeletal muscles [64,65]. In agreement with previous studies [66,67,68], in the present study, no changes in the expression of mitochondrial biogenesis associated genes in the skeletal muscle of SirT1 OE mice. Overall, these results reveal that SirT1 OE does not affect whole-body energy expenditure or augment skeletal muscle insulin sensitivity regardless of aging in response to AL diet, which is similar to studies from other labs [66,68,69]. It is important to admit that germline OE of SirT1 typically elevates SirT1 levels several-fold and such large-scale OE may preempt changes that are beneficial to metabolic action. Notably, however, Banks et al. [70] showed that two- to three-fold SirT1 OE from germline, which could be considered as a physiological level, did not alter muscle insulin action. It may be valuable to use SirT1 pharmacological activator to activate SirT1 in SirT1 OE mice in vivo to enhance its function. In our study, we used global SirT1 OE mice. SirT1 OE exclusively in the brain demonstrated a decreased whole-body energy expenditure, and concomitantly a reduction in the expression of mitochondria-related genes in muscle [71]. In addition, these mice showed unusual spontaneous physical activity patterns, decreased activity in the open field, and rotarod performance. Furthermore, aging is associated with increased apoptosis in skeletal muscle. SirT1 OE in the present study inhibited pathways linked to apoptosis [72,73]. This may be mediated through the well-established downstream target of SirT1 p53, which inhibition by SirT1-mediated deacetylation-dependent mechanism reduces apoptosis and oxidative damage [74]. Interestingly, SirT1 OE activated GTPase signaling in aging muscle. The Rho GTPases Rac1, RhoA, and Cdc42 are highly expressed in skeletal muscle and emerging evidence suggests that they have key functions in the regulation of glucose uptake and the maintenance of whole-body glucose homeostasis [75,76,77]. These data suggest that SirT1 OE may reverse age-associated repressed glucose homeostasis as observed in WT mice.

To study the effects of CR on the skeletal muscle gene expression profile during aging, we first compared the skeletal muscle RNA-seq data of aged WT mice fed to AL or CR diet. DGE analysis identified that while aging did not alter many genes in the skeletal muscle of WT mice compared to young mice fed the AL diet, more genes (339 genes) are responded to CR in aged WT mice. However, in young mice CR altered more genes in young mice (802 genes) than aged mice. This indicates that aging diminishes global gene transcriptional rates, which was reversed by CR. The genes that responded to CR in aged mice are associated with cellular transcription and translation such as mRNA, ncRNA, and rRNA processing, mRNA splicing, cytoplasmic translation, RNA binding, and ribosomal biogenesis. These results imply that CR may reverse the age-induced loss of skeletal muscle gene expression profile by increasing the pathways controlling cellular gene transcriptional and translational processes. Regardless of aging, most of the CR-altered genes are downregulated in skeletal muscle. Interestingly, the CR responsive genes related to cellular transcription and translation are repressed in young mice, suggesting that CR has a contradictory role between young and aged mice. This opposing role may be linked to preserving some of the functions of CR in aging muscle. The pathways that are hindered in aged muscles following CR are connected to ECM and cytoskeleton such as collagen fibril organization, focal adhesion, and integrin binding. However, CR inhibited more pathways of ECM and cytoskeletal in young mice, signifying a role for CR in negatively regulating ECM and cytoskeletal pathways irrespective of aging.

To study the role of SirT1 in the regulation of genes that are coordinated by CR on the skeletal muscle during aging, we compared the skeletal muscle RNA-seq data of young and aged SirT1 KO and OE mice fed to AL or CR diet. DGE analysis identified that aged SirT1 KO mice fed to CR displayed fewer differentially expressed genes (465 genes) compared to young SirT1 KO mice (1296 genes), which is similar to aged KO mice fed AL diet. In contrast, CR altered more genes (1465) in aged SirT1 OE mice compared to young SirT1 OE mice (994 genes), suggesting a role for SirT1 in controlling genome-wide gene expression profile. Most of the genes that responded to CR are downregulated regardless of aging and phenotypes. Interestingly, the suppressed genes that responded to CR in aged SirT1 KO mice were associated with cellular transcription and translation, which are opposite to CR fed aged WT and SirT1 OE mice in which ECM and cytoskeletal protein (WT), and muscle structure and contraction associated pathways (SirT1 OE) are repressed. This suggests that SirT1 and CR have different roles in the regulation of muscle biological pathways. In aged SirT1 KO mice, CR upregulated fatty acid oxidation and mitochondrial function-related pathways. This alteration is contradictory to young SirT1 KO mice in which the most abundant downregulated pathways in response to CR were related to the mitochondrial electron transport chain. Some of these pathways are also suppressed in aged SirT1 KO mice fed an AL diet. Interestingly, young SirT1 KO mice fed the AL diet showed an upregulation of pathways associated with mitochondrial oxidative phosphorylation. This indicates that CR likely preserves mitochondrial function in aged SirT1 KO mice. In SirT1 KO mice, more muscle contractile proteins are upregulated in young mice compared to aged mice whereas CR restored these proteins in aged SirT1 KO mice along with fatty acid β-oxidation. This suggests that CR may improve muscle function during aging and SirT1 is not essential for such progress. Interestingly, pathways associated with muscle contractile apparatus are highly suppressed in response to CR in SirT1 OE mice. Our data suggest that SirT1 may have a negative role in the regulation of muscle contractile proteins. This is further confirmed in young SirT1 OE mice, in which CR downregulated ECM and muscle contractile apparatus pathways.

## 5. Conclusions

The results from our RNA sequencing study indicate wide-scale transcriptome alterations in skeletal muscle following SirT1 loss-of-function or CR in both young and aged mice. Loss of insulin signaling and muscle activity, as well as mitochondrial function-related pathways, were observed in SirT1 KO mice fed the AL diet. SirT1 OE does not affect whole-body energy expenditure or augment skeletal muscle insulin sensitivity regardless of aging in response to the AL diet. CR reversed the age-induced loss of skeletal muscle gene expression profile by increasing the pathways controlling cellular gene transcriptional and translational processes. CR negatively regulates ECM and cytoskeletal pathways irrespective of aging and preserves mitochondrial function in aged SirT1 KO mice. CR may improve muscle function during aging and SirT1 is not essential for such improvement. Considering the complex, multi-faceted mechanisms driving CR-induced skeletal muscle gene expression profile, these results provide preliminary investigative guidance. Therefore, it may be prudent to direct future research toward the pathways associated with the DE genes identified in the current study.

## Figures and Tables

**Figure 1 genes-12-00691-f001:**
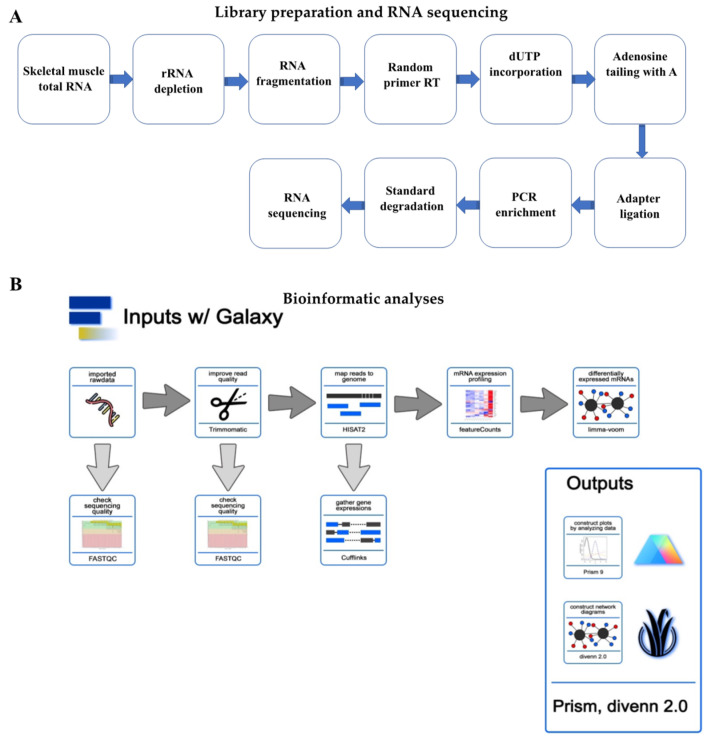
A brief workflow of library preparation (**A**) and RNA sequencing (**B**) performed in the present study.

**Figure 2 genes-12-00691-f002:**
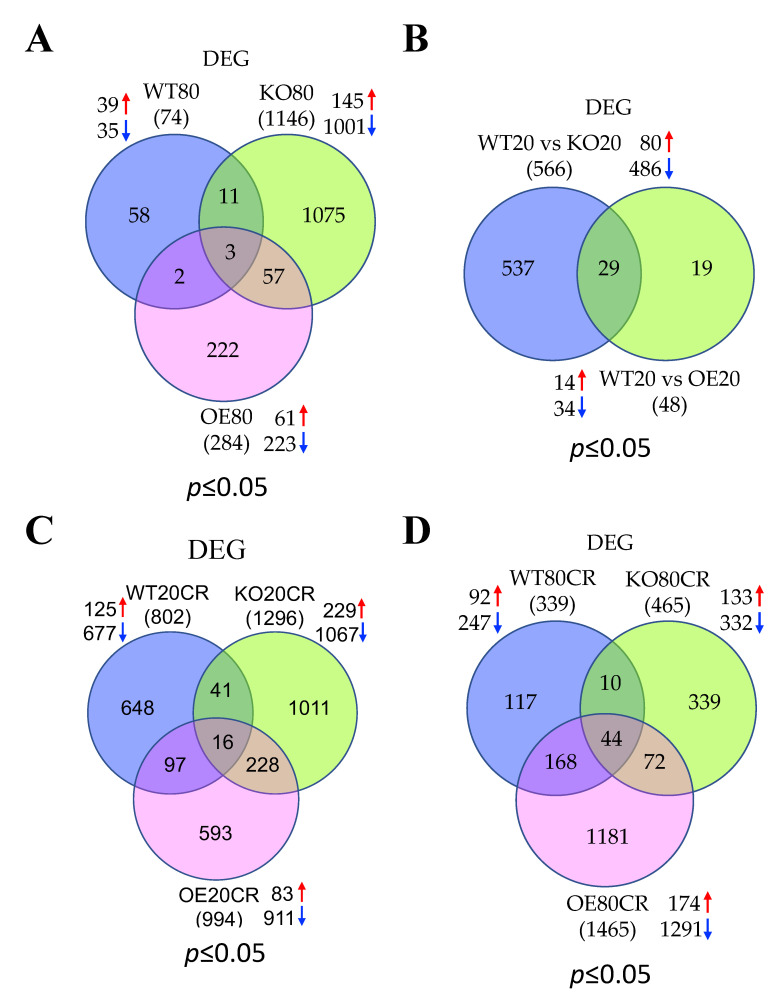
Venn diagrams display the comparison of differentially expressed genes (DEG) between groups as follows: (**A**) Aged WT vs. SirT1 KO vs. SirT1 OE mice; (**B**) young WT vs. SirT1 KO or SirT1 OE mice; (**C**) young WT vs. SirT1 KO vs. SirT1 OE mice in response to CR; (**D**) Aged WT vs. SirT1 KO vs. SirT1 OE mice in response to CR. Numbers inside the parenthesis indicate total number of differentially regulated genes whereas the numbers inside Venn diagrams indicate total number of differentially regulated genes unique to each group. Red and blue arrows indicate up- and downregulated genes, respectively. WT: wild-type; KO: knockout; OE: overexpression.

**Figure 3 genes-12-00691-f003:**
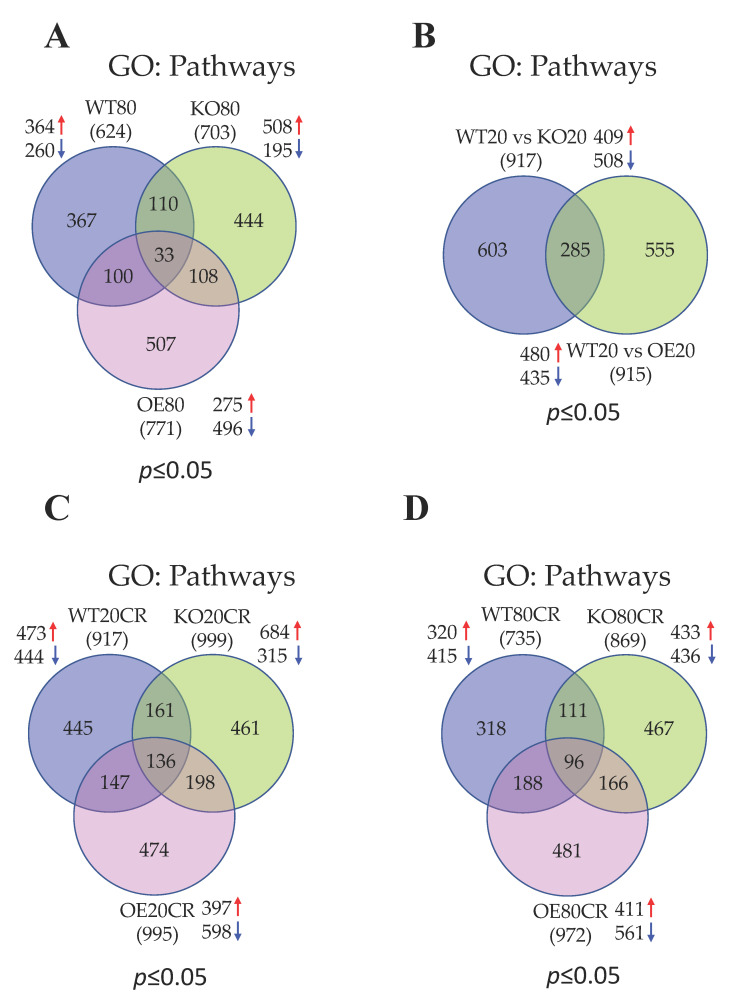
Venn diagrams show the comparison of number of significantly regulated biological pathways between groups as follows: (**A**) Aged WT vs. SirT1 KO vs. SirT1 OE mice; (**B**) young WT vs. SirT1 KO or SirT1 OE mice; (**C**) young WT vs. SirT1 KO vs. SirT1 OE mice in response to CR; (**D**) Aged WT vs. SirT1 KO vs. SirT1 OE mice in response to CR. Numbers inside the parenthesis indicate total number of differentially regulated pathways whereas the numbers inside Venn diagrams indicate total number of differentially regulated pathways unique to each group. Red and blue arrows indicate up- and downregulated genes. GO: go term; WT: wild-type; KO: knockout; OE: overexpression.

**Table 1 genes-12-00691-t001:** Project information.

**Sample Information**												
Sample ID	WT20AL	WT20CR	WT80AL	WT80CR	KO20AL	KO20CR	KO80AL	KO80CR	OE20AL	OE20CR	OE80AL	OE80CR
Sample size	4	4	4	4	4	4	4	4	4	4	4	4
**Database**												
Genome assembly		https://useast.ensembl.org/Mus_musculus/Info/Index: GRCm38.p6 (GCA_000001635.8)										
Gene annotation		ftp://ftp.ensembl.org/pub/release-102/gtf/mus_musculus//Mus_musculus.GRCm38.102.gtf.gz										
lncRNA		ftp://ftp.sanger.ac.uk/pub/gencode/Gencode_mouse/release_M13/gencode.vM13.long_noncoding_RNAs.gtf.gz (v13)										
Gene Orthology (GO)		http://www.geneontology.org/ (accessed on 1 January 2021)										
KEGG		http://www.genome.jp/kegg/pathway.html (accessed on 18 January 2021)										
**Bioinformatics software**												
Quality control		FastQC (vs. 0.11.9)										
Adaptors remove		Trimmomatic (vs. 0.39)										
Mapping		HISAT2 (vs. 2.1.0)										
Transcripts assembly		StringTie (vs. 2.1.1)										
Differential expression analysis		Limma (vs. 3.42)										
GO and KEGG enrichment analysis		g:Profiler version e102_eg49_p15_7a9b4d6										
Coding potential		CPC (Coding Potential Calculator) (vs. 0.9)										
Coding potential		CNCI (Coding-Non-Coding Index) (vs. 2.0)										

**Table 2 genes-12-00691-t002:** A brief comparison of total number of differentially expressed genes (*p* ≤ 0.05) and associated essential pathways.

Mice	Age	Diet	Genes	Up	Down	Upregulated Pathways	Downregulated Pathways
WT	O	AL	74	39	35	Muscle contraction; Cellular transcription and translation	Ubiquitin; Energy homeostasis
KO	O	AL	1146	145	1001	Cellular Transcriptions	Mitochondrial functions
OE	O	AL	284	61	223	GTPase signaling; DNA binding activity; Ion channel functions	Apoptotic signaling; Metabolism
KO	Y	AL	566	80	486	Mitochondrial functions; Metabolism; Ubiquitin activity	Muscle function; Insulin signaling
OE	Y	AL	48	14	34	Signal transduction; Transcription corepressor activity	Lipid metabolism
WT	Y	CR	802	125	677	Phosphatidylinositol binding	Translational; Muscle contraction
KO	Y	CR	1296	229	1067	Muscle contraction	Mitochondrial function
OE	Y	CR	994	83	911	Gene expression	ECM; Cytoskeletal proteins
WT	O	CR	339	92	247	Transcription and translation	ECM; Cytoskeletal proteins
KO	O	CR	465	133	332	Mitochondrial β-oxidation; fatty acid metabolism; muscle contraction	Cellular gene expression
OE	O	CR	1465	174	1291	Ubiquitination signaling	Muscle structure; Contraction

**Table 3 genes-12-00691-t003:** Differentially regulated genes (≥2-fold and *p* ≤ 0.05) in aged WT, and SirT1 KO and OE mice fed to AL diet.

No.	Genes	logFC	Adj.P.Val	No.	Genes	logFC	Adj.P.Val
*WT80*				2	*Leng8*	−2.25	0.05
1	*Gm6472*	3.10	0.00	3	*Sppl2a*	−2.21	0.05
2	*Rps13-ps1*	2.85	0.00	4	*Arid1b*	−2.21	0.05
3	*Gm15920*	2.81	0.00	5	*Agpat1*	−2.16	0.05
4	*Ngp*	2.50	0.01	6	*Dnajc18*	−2.14	0.04
5	*Ltf*	2.41	0.00	7	*Rock2*	−2.10	0.05
6	*S100a8*	2.41	0.03	8	*Fto*	−2.09	0.04
7	*S100a9*	2.38	0.03	9	*Atp6v0a2*	−2.09	0.04
8	*Igkc*	2.33	0.00	10	*St13*	−2.08	0.05
9	*Gm9855*	2.20	0.00	11	*Plin4*	−2.08	0.05
10	*Chil3*	2.04	0.05	12	*Crk*	−2.08	0.04
*KO80*				13	*Cbx5*	-2.07	0.03
1	*Gm5537*	2.87	0.00	14	*Baz2a*	−2.05	0.04
2	*Hmgb1-ps8*	2.71	0.00	15	*Phc3*	−2.03	0.04
3	*Lcn2*	2.53	0.00	16	*Fbxo38*	−2.02	0.03
4	*Gm9794*	2.40	0.02	17	*Wfikkn2*	−2.02	0.05
5	*Pakap*	2.15	0.03	18	*St3gal3*	−2.01	0.04
6	*Mt2*	2.12	0.02	19	*Slc9a8*	−2.01	0.04
7	*Igkc*	2.06	0.02	20	*AU022252*	−2.01	0.05
*OE80*				21	*Erap1*	-2.00	0.05
1	*Gm28438*	2.83	0.04	22	*Ckap5*	−2.00	0.04
2	*Mup10*	2.69	0.00	23	*Mlh3*	−2.00	0.04
3	*Myo15*	2.53	0.00	24	*Ccdc97*	−2.00	0.03
*KO80*							
1	*Zfp148*	−2.27	0.05				

**Table 4 genes-12-00691-t004:** Differentially regulated genes (≥2-fold and *p* ≤ 0.05) in young SirT1 KO and OE mice fed to AL diet.

No.	Genes	log2FC	Adj.P.Val	No.	Genes	log2FC	Adj.P.Val
*SirT1 KO*				26	*Cd81*	−2.15	0.04
1	*Gm15920*	3.09	0.00	27	*Atp5j*	−2.13	0.03
2	*Rps13-ps1*	2.93	0.00	28	*Prpf38b*	−2.11	0.05
3	*Gm15459*	2.88	0.00	29	*Sec62*	−2.10	0.03
4	*Gm2539*	2.69	0.00	30	*Atp5h*	−2.10	0.05
5	*Gm19791*	2.68	0.00	31	*Mrpl18*	−2.09	0.03
6	*Rps12l1*	2.53	0.02	32	*Dnajc21*	−2.09	0.05
7	*Gm7993*	2.44	0.00	33	*Eif2b4*	−2.08	0.05
8	*Mrpl48-ps*	2.32	0.00	34	*Cbfb*	−2.08	0.04
9	*Gm9855*	2.31	0.00	35	*Ndufs3*	−2.07	0.04
10	*Gm14586*	2.28	0.01	36	*Ewsr1*	−2.06	0.03
11	*Rtkn2*	2.03	0.00	37	*Serf2*	−2.06	0.02
12	*Ighm*	2.00	0.00	38	*Six1*	−2.05	0.03
13	*Vdac3*	−2.40	0.05	39	*Tceb1*	−2.05	0.02
14	*Cisd2*	−2.35	0.05	40	*Aplp2*	−2.04	0.02
15	*Ndufs6*	−2.28	0.04	41	*Gpx4*	−2.03	0.02
16	*Rps25*	−2.24	0.05	42	*Osgepl1*	−2.03	0.04
17	*Hnrnpa2b1*	−2.21	0.04	43	*Fxr1*	−2.03	0.02
18	*Cox7c*	−2.21	0.03	44	*Rps14*	−2.02	0.05
19	*Rhoa*	−2.21	0.04	45	*Ppp2r4*	−2.01	0.04
20	*Maf*	−2.18	0.05	46	*Thumpd1*	−2.01	0.03
21	*Ndufb7*	−2.18	0.03	47	*Zmat2*	−2.01	0.04
22	*Pkia*	−2.18	0.04	48	*Uba3*	−2.01	0.04
23	*Psmd12*	−2.16	0.04	49	*Thumpd1*	−2.15	0.04
24	*Mtch2*	−2.15	0.03	50	*Zmat2*	−2.13	0.03
25	*Apobec2*	−2.15	0.05	51	*Uba3*	−2.11	0.05

**Table 5 genes-12-00691-t005:** Differentially upregulated genes (≥2-fold and *p* ≤ 0.05) in response to CR in young and aged WT, and SirT1 KO and OE mice.

No.	Genes	log2FC	Adj. P. Val	No.	Genes	log2FC	Adj. P. Val
*WT20*				*KO80*			
1	*Ddx3y*	3.25	0.00	1	*Gm6472*	0.01	2.69
2	*Eif2s3y*	3.01	0.00	2	*Padi1*	0.01	2.14
3	*Rpl3-ps1*	2.81	0.00	*OE80*			
4	*Slc15a5*	2.45	0.00	1	*Eif2s3y*	0.00	3.01
5	*Uty*	2.30	0.00	2	*Ddx3y*	0.00	2.99
6	*Vmn1r65*	2.15	0.00	3	*Gm12191*	0.01	2.47
*KO20*				4	*Slc15a5*	0.00	2.42
1	*Gm5537*	2.93	0.00	5	*Uty*	0.00	2.40
2	*Igha*	2.44	0.00	6	*Kcnf1*	0.00	2.29
3	*Ddit4*	2.14	0.00	7	*Sult1e1*	0.00	2.17
4	*Gm26992*	2.08	0.00	8	*RP24-258B3.4*	0.00	2.03
5	*Gm9794*	2.05	0.04	9	*C7*	0.00	2.00
6	*Lcn2*	2.03	0.00	10	*1700001O22Rik*	0.00	2.00
*OE20*							
1	*Doc2b*	0.00	2.22				
2	*Gm21541*	0.02	2.13				

**Table 6 genes-12-00691-t006:** Differentially downregulated genes (≥2-fold and *p* ≤ 0.05) in response to CR in young WT, and SirT1 KO and OE mice.

No.	Genes	log2FC	Adj. P. Val	No.	Genes	log2FC	Adj. P. Val
*WT20*				41	*Dnajc27*	0.04	−2.00
1	*Fth1*	0.05	−2.43	42	*Trappc8*	0.05	−2.00
2	*Ank*	0.05	−2.40	*KO20*			
3	*Unc45b*	0.05	−2.38	1	*Tnrc6b*	0.04	−2.27
4	*Ehbp1l1*	0.05	−2.35	2	*Usp24*	0.04	−2.26
5	*Rnf11*	0.05	−2.31	3	*Prpf4*	0.05	−2.19
6	*Rxra*	0.05	−2.29	4	*Got2*	0.04	−2.18
7	*Kif1b*	0.04	−2.29	5	*Thrap3*	0.04	−2.16
8	*Srl*	0.03	−2.28	6	*Ralgapb*	0.05	−2.15
9	*Ankrd40*	0.04	−2.25	7	*Ppp2ca*	0.05	−2.15
10	*Klhl24*	0.05	−2.23	8	*Bptf*	0.05	−2.15
11	*Sbno1*	0.05	−2.23	9	*Camsap1*	0.05	−2.14
12	*Setd7*	0.05	−2.22	10	*Osbp*	0.04	−2.14
13	*Ap2a1*	0.05	−2.21	11	*Epb41*	0.03	−2.14
14	*Cds2*	0.05	−2.20	12	*Zer1*	0.05	−2.13
15	*Tmem64*	0.03	−2.18	13	*Eif4g3*	0.04	−2.10
16	*Wwp1*	0.03	−2.17	14	*Asb11*	0.04	−2.07
17	*Otud4*	0.05	−2.17	15	*Cep104*	0.04	−2.06
18	*Ppp2r2a*	0.05	−2.16	16	*Echs1*	0.03	−2.06
19	*Wdtc1*	0.05	−2.16	17	*Dnajc13*	0.04	−2.06
20	*Ube2n*	0.05	−2.15	18	*Ptpn1*	0.05	−2.05
21	*Cyhr1*	0.04	−2.15	19	*Pum1*	0.04	−2.05
22	*Nipbl*	0.05	−2.13	20	*Ap3d1*	0.05	−2.05
23	*Pdpr*	0.02	−2.11	21	*Snx17*	0.03	−2.04
24	*Pdzrn3*	0.05	−2.10	22	*Eif4g1*	0.03	−2.04
25	*Ulk1*	0.04	−2.09	23	*Lrch3*	0.05	−2.04
26	*Usp13*	0.02	−2.08	24	*Ubap2l*	0.05	−2.04
27	*Isca1*	0.02	−2.06	25	*Aldh6a1*	0.02	−2.01
28	*Pja2*	0.05	−2.05	26	*Tceb3*	0.04	−2.01
29	*Vwa8*	0.03	−2.05	*OE20*			
30	*Tnrc6a*	0.03	−2.04	1	*Tcp1*	0.05	−2.36
31	*Mtus1*	0.04	−2.04	2	*Arhgef6*	0.05	−2.35
32	*Ergic2*	0.05	−2.03	3	*Mbnl2*	0.04	−2.33
33	*Tubb4b*	0.03	−2.02	4	*Rplp2*	0.04	−2.29
34	*Tbc1d16*	0.05	−2.02	5	*Jtb*	0.05	−2.28
35	*Timp3*	0.05	−2.01	6	*Synrg*	0.04	−2.27
37	*Fbxo11*	0.05	−2.01	7	*Apoo*	0.05	−2.26
36	*Dhx32*	0.03	−2.01	8	*Ctsl*	0.05	−2.25
37	*Ryk*	0.04	−2.00	9	*Rps15*	0.04	−2.25
38	*Sf3b2*	0.04	−2.24	10	*Pdk2*	0.03	−2.09
39	*Dlg1*	0.04	−2.00	11	*Dnaja2*	0.04	−2.23
12	*Nrd1*	0.04	−2.22	46	*Eprs*	0.04	−2.08
13	*Ppp6c*	0.05	−2.22	47	*Phf2*	0.04	−2.08
14	*Rps18*	0.04	−2.22	48	*Fbxw2*	0.03	−2.08
15	*Tex2*	0.05	−2.20	49	*Dnajc7*	0.05	−2.08
16	*Luc7l2*	0.05	−2.20	50	*Ndel1*	0.03	−2.08
17	*Mtdh*	0.04	−2.19	51	*Pnn*	0.04	−2.07
18	*Rpl35*	0.04	−2.17	52	*Csde1*	0.03	−2.07
19	*Atp2b1*	0.03	−2.17	53	*Ddx1*	0.03	−2.07
20	*Rab18*	0.05	−2.17	54	*Atp8a1*	0.04	−2.07
21	*Acacb*	0.04	−2.17	55	*Nap1l1*	0.03	−2.06
22	*Zbtb18*	0.05	−2.16	56	*Rnf150*	0.04	−2.06
23	*Ppp1r2*	0.03	−2.16	57	*Rpl23*	0.03	−2.06
24	*Usp15*	0.03	−2.16	58	*Ppp3cb*	0.05	−2.06
25	*Serinc1*	0.03	−2.16	59	*Msrb1*	0.04	−2.06
26	*Agpat3*	0.04	−2.16	60	*Rps6*	0.02	−2.06
27	*Zfand3*	0.04	−2.15	61	*Mylk2*	0.04	−2.05
28	*Zc3h14*	0.05	−2.15	62	*Sgca*	0.05	−2.05
29	*Rpl15*	0.03	−2.15	63	*Rps11*	0.05	−2.05
30	*Atf7*	0.04	−2.14	64	*Rap1gds1*	0.03	−2.04
31	*Atp6v1g1*	0.03	−2.14	65	*Scaf1*	0.05	−2.04
32	*Dcaf6*	0.05	−2.13	66	*Fem1a*	0.04	−2.04
33	*Homer1*	0.04	−2.12	67	*Pfdn5*	0.02	−2.03
34	*Smad4*	0.05	−2.12	68	*Bmi1*	0.02	−2.03
35	*Fgfr1*	0.04	−2.12	69	*Ccar1*	0.04	−2.03
36	*Set*	0.03	−2.11	70	*Abhd16a*	0.04	−2.03
37	*Nsmaf*	0.05	−2.11	71	*Golga3*	0.03	−2.03
38	*Impad1*	0.05	−2.11	72	*Arf1*	0.05	−2.02
39	*Sugt1*	0.04	−2.11	73	*Fyco1*	0.02	−2.02
40	*Scn4a*	0.05	−2.10	74	*Prkcq*	0.02	−2.02
41	*Zfand5*	0.03	−2.10	75	*Wfdc1*	0.03	−2.02
42	*Map1lc3b*	0.04	−2.10	76	*Ubxn4*	0.03	−2.00
43	*Vps13d*	0.05	−2.10	77	*Drap1*	0.05	−2.00
45	*Adipor1*	0.04	−2.09				

**Table 7 genes-12-00691-t007:** Differentially expressed genes (≥2-fold and *p* ≤ 0.05) in response to CR in 80 wk old WT, SirT1 KO and OE mice.

No.	Genes	logFC	Adj.P.Val	No.	Genes	logFC	Adj.P.Val
*KO80*				26	*Ltn1*	0.05	−2.40
1	*Gm6472*	2.69	0.01	27	*Ipo7*	0.04	−2.39
2	*Padi1*	2.14	0.01	28	*Btbd1*	0.02	−2.39
*OE80*				29	*Snx5*	0.03	−2.39
1	*Eif2s3y*	3.01	0.00	30	*Atf7ip*	0.05	−2.38
2	*Ddx3y*	2.99	0.00	31	*Srp72*	0.04	−2.37
3	*Gm12191*	2.47	0.01	32	*Araf*	0.05	−2.36
4	*Slc15a5*	2.42	0.00	33	*Rnf10*	0.03	−2.35
5	*Uty*	2.40	0.00	34	*Tmed2*	0.05	−2.35
6	*Kcnf1*	2.29	0.00	35	*Jph1*	0.03	−2.35
7	*Sult1e1*	2.17	0.00	36	*Lmtk2*	0.05	−2.35
8	*RP24-258B3.4*	2.03	0.00	37	*Obscn*	0.04	−2.35
9	C7	2.00	0.00	38	*Ncor1*	0.03	−2.34
10	*1700001O22Rik*	2.00	0.00	39	*Dhx15*	0.04	−2.34
*OE80*				40	*Mtmr3*	0.03	−2.33
1	*Tsnax*	0.05	−2.64	41	*Map2k7*	0.03	−2.32
2	*Fto*	0.05	−2.62	42	*Eif4g1*	0.03	−2.32
3	*Matr3*	0.05	−2.60	43	*Eif2b5*	0.05	−2.32
4	*Ywhag*	0.05	−2.59	44	*Actr1a*	0.05	−2.31
5	*Hdlbp*	0.04	−2.58	45	*Adipor2*	0.03	−2.30
6	*Setd3*	0.05	−2.56	46	*Tmbim6*	0.02	−2.30
7	*Map3k7*	0.05	−2.53	47	*Insr*	0.03	−2.29
8	*Tmem38a*	0.05	−2.51	48	*Ufl1*	0.04	−2.29
9	*Myh4*	0.03	−2.51	49	*Dctn2*	0.02	−2.29
10	*Polr1d*	0.05	−2.51	50	*Rad23b*	0.03	−2.28
11	*Rsl1d1*	0.05	−2.50	51	*Cnot10*	0.05	−2.28
12	*Trpc4ap*	0.05	−2.50	52	*Cnot1*	0.04	−2.28
13	*Clint1*	0.05	−2.50	53	*Isca1*	0.04	−2.28
14	*Spag7*	0.04	−2.50	54	*Sumo1*	0.02	−2.28
15	*Maf1*	0.05	−2.49	55	*Arglu1*	0.05	−2.28
16	*Nudt3*	0.04	−2.47	56	*Dusp27*	0.02	−2.27
17	*Ddx6*	0.04	−2.46	57	*Eif2a*	0.03	−2.26
18	*Dlg1*	0.02	−2.46	58	*Sox6*	0.05	−2.26
19	*Rfx7*	0.04	−2.45	59	*Cnbp*	0.05	−2.26
20	*Chd4*	0.02	−2.43	60	*Tom1*	0.03	−2.26
21	*Rfk*	0.02	−2.42	61	*Atg4a*	0.04	−2.26
22	*Wdtc1*	0.05	−2.41	62	*Usp4*	0.03	−2.25
23	*Ar*	0.05	−2.41	63	*Ei24*	0.02	−2.25
24	*Stau1*	0.03	−2.41	64	*Ccdc43*	0.03	−2.25
25	*Akt2*	0.03	−2.41	62	*Eif3l*	0.05	−2.25
66	*Pitpnb*	0.03	−2.24	111	*Klf9*	0.01	−2.11
67	*Paip2b*	0.03	−2.24	112	*Eif3j2*	0.03	−2.11
68	*Gapvd1*	0.04	−2.24	113	*Cds2*	0.02	−2.11
69	*Synrg*	0.01	−2.23	114	*Trip12*	0.01	−2.11
70	*Rab3gap2*	0.05	−2.23	115	*Eif1*	0.05	−2.11
71	*Usp7*	0.03	−2.22	116	*Cap2*	0.04	−2.11
72	*Eif4a1*	0.04	−2.21	117	*Chmp4b*	0.02	−2.11
73	*Mfap1b*	0.02	−2.21	118	*Srsf10*	0.04	−2.10
74	*Pcyt1a*	0.01	−2.20	119	*Atp6v1c1*	0.02	−2.10
75	*Adipor1*	0.03	−2.20	120	*Atmin*	0.03	−2.10
76	*Tcap*	0.02	−2.20	121	*Phkb*	0.02	−2.10
77	*Pacsin3*	0.05	−2.19	122	*Arnt*	0.04	−2.09
78	*Kat7*	0.03	−2.18	123	*Mbtps1*	0.02	−2.09
79	*Ddx5*	0.04	−2.18	124	*Dnajb5*	0.01	−2.09
80	*Chmp1a*	0.03	−2.18	125	*Kpna3*	0.01	−2.09
81	*Nphp1*	0.03	−2.18	126	*Ttc9c*	0.05	−2.09
82	*Txnl1*	0.04	−2.17	127	*Sympk*	0.02	−2.09
83	*Pdxdc1*	0.03	−2.17	128	*Txndc16*	0.05	−2.09
84	*Axin1*	0.05	−2.17	129	*Nup98*	0.05	−2.09
85	*Usp15*	0.03	−2.17	130	*Dhx29*	0.01	−2.09
86	*Wac*	0.05	−2.16	131	*Ccni*	0.01	−2.09
87	*Marf1*	0.05	−2.16	132	*Ttc3*	0.02	−2.09
88	*Tmem214*	0.05	−2.16	133	*Psmd11*	0.03	−2.08
89	*Bcap29*	0.03	−2.16	134	*Limd1*	0.01	−2.08
90	*E430025E21Rik*	0.04	−2.16	135	*Plaa*	0.03	−2.08
91	*Det1*	0.05	−2.16	136	*Mybpc2*	0.02	−2.08
92	*Oxa1l*	0.03	−2.15	137	*Dedd*	0.05	−2.08
93	*Mrps30*	0.05	−2.15	138	*Zbtb18*	0.03	−2.08
94	*Eif2b2*	0.05	−2.15	139	*Ubl7*	0.05	−2.07
95	*Prpf4b*	0.03	−2.15	140	*Hsp90ab1*	0.03	−2.07
96	*Fbrs*	0.05	−2.14	141	*Sra1*	0.04	−2.07
97	*Son*	0.01	−2.14	142	*Rab3gap1*	0.04	−2.07
98	*Psmb1*	0.02	−2.14	143	*Sgta*	0.02	−2.07
99	*2310036O22Rik*	0.04	−2.14	144	*Smad4*	0.05	−2.07
100	*Mtor*	0.03	−2.14	145	*Ift172*	0.02	−2.06
101	*Tmem30a*	0.02	−2.14	146	*Csde1*	0.04	−2.06
102	*0610009O20Rik*	0.03	−2.13	147	*Fam63a*	0.03	−2.06
103	*Polr2a*	0.01	−2.13	148	*Arhgap35*	0.04	−2.06
104	*Psmc3*	0.02	−2.13	149	*Phrf1*	0.05	−2.06
105	*Zc3h4*	0.05	−2.13	150	*Kif1bp*	0.02	−2.06
106	*Mgat4b*	0.05	−2.13	151	*Abcf1*	0.02	−2.06
107	*Cab39*	0.05	−2.13	152	*Tm9sf2*	0.03	−2.06
108	*Ppfia1*	0.04	−2.12	153	*Cep120*	0.04	−2.05
109	*Camta2*	0.01	−2.12	154	*Clptm1*	0.03	−2.05
110	*Syap1*	0.03	−2.12	155	*Psmd3*	0.02	−2.05
156	*Thrap3*	0.01	−2.05	174	*1110037F02Rik*	0.02	−2.02
157	*Cnot4*	0.04	−2.05	175	*Ubqln2*	0.05	−2.02
158	*Sh3pxd2a*	0.02	−2.04	176	*Phf14*	0.02	−2.02
159	*Stx12*	0.01	−2.04	177	*Akap9*	0.02	−2.02
160	*Rac1*	0.01	−2.04	178	*Per2*	0.04	−2.02
161	*Npepl1*	0.03	−2.04	179	*Rbm5*	0.03	−2.02
162	*Tsc1*	0.04	−2.04	180	*Ewsr1*	0.01	−2.01
163	*Dnajc2*	0.04	−2.04	181	*Gcn1l1*	0.01	−2.01
164	*Gpx4*	0.03	−2.04	182	*Fryl*	0.02	−2.01
165	*Crot*	0.05	−2.03	183	*Eea1*	0.04	−2.01
166	*Msrb1*	0.05	−2.03	184	*Las1l*	0.02	−2.01
167	*Ikzf5*	0.02	−2.03	185	*Dnajb11*	0.01	−2.01
168	*Ulk1*	0.03	−2.03	186	*Tpd52l1*	0.04	−2.00
169	*Glg1*	0.01	−2.03	187	*Psmc2*	0.03	−2.00
170	*Crbn*	0.01	−2.03	188	*Gna11*	0.02	−2.00
171	*Ubr1*	0.04	−2.03	189	*Osgep*	0.05	−2.00
172	*Ubxn6*	0.02	−2.02	190	*Tmem43*	0.02	−2.00
173	*Sdf4*	0.02	−2.02				

## Data Availability

Not applicable.

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
