# Peer review of "Skeletal Muscle Gene Expression Profile in Response to Caloric Restriction and Aging: A Role for SirT1"

_genes, 2021, doi:10.3390/genes12050691_

Round 1
Reviewer 1 Report
In the manuscript by Myers et.al., the authors investigate the role of SirT1 in regulating transcription expression profiles of muscles in the calorie restricted mice model of aging. The authors have performed the RNA-Seq profiling of muscles from the 20-week and 80-week-old wild-type, SirT1 knock-out, and SirT1 over-expression mice upon caloric restriction. The manuscript is data intensive and authors have performed multiple comparative analysis. Although the question addressed in this work is interesting, the manuscript lacks the flow, description, and depth in the analysis. Many of the data shown in main figures is the quality control data and can be either put in supplementary or not included at all. The details of the analysis should be inserted in figure legends and table legends.
The major comments to be considered to improve the quality of the manuscript are:
- The authors should only focus on statistically significant expression changes and should exclude the list of differentially expressed genes based on expression from the manuscript.
- The details of comparative groups should be included in figure legends and table legends. They need to be more information intensive.
- The authors should also perform analysis to see comparative significant changes that show a pattern between SirT1 over-expression, WT and SirT1 knockout muscles. Also, does these changes are altered in calorie restricted data in comparison to normal diet.
- The focus of the manuscript should be highlighting SirT1-regulated changes that differ in calorie-restricted young and old mice. It is not reflected properly in the result section of the manuscript.
- Many of the data is QC data and be put in supplementary or removed from the manuscript. Fig2, 3, and 5.
- For Fig4, it will be better to show heatmap of significant DE genes between different groups.
- Fig6 and Fig7 are not Venn diagrams as written in figure legends.
- It will be better to include a table showing all the significantly up- and down-regulated genes between different groups.
Author Response
We sincerely thank the reviewers for their great suggestions. We have significantly improved the original manuscript by reanalyzing the data. We removed odd and bad quality RNA-seq data from the previous analysis. In the revised version, we have used only 4 good RNA-seq paired-end reads from each group (n=4). As a result, we now have better and interesting as well as a meaningful full outcome.
The authors should only focus on statistically significant expression changes and should exclude the list of differentially expressed genes based on expression from the manuscript.
We have selected and used only statistically significant data to identify their biological significance in the revised manuscript.
The details of comparative groups should be included in figure legends and table legends. They need to be more information intensive.
Our new figure captions have all these components.
The authors should also perform analysis to see comparative significant changes that show a pattern between SirT1 over-expression, WT and SirT1 knockout muscles. Also, does these changes are altered in calorie restricted data in comparison to normal diet.
The result and discussion part in the revised manuscript has been focused to discuss only the role of SirT1 in the regulation of CR responsive gene expression and associated pathways.
The focus of the manuscript should be highlighting SirT1-regulated changes that differ in calorie-restricted young and old mice. It is not reflected properly in the result section of the manuscript.
We have seriously considered these points, which were well-taken in the revised manuscript.
Many of the data is QC data and be put in supplementary or removed from the manuscript. Fig2, 3, and 5.
The revised manuscript does not have the QC data.
For Fig4, it will be better to show a heatmap of significant DE genes between different groups.
We have removed the heatmap and included volcano plots in the revised manuscript.
Fig6 and Fig7 are not Venn diagrams as written in figure legends.
This has been fixed by including new Venn diagrams.
It will be better to include a table showing all the significantly up-and down-regulated genes between different groups.
We have included new tables showing all the significantly up-and down-regulated genes between different groups.
Reviewer 2 Report
The paper "Skeletal Muscle Gene Expression Profile In Response to Ca-2 loric Restriction During Aging: A role for SirT1" by Myers et al. suggests the role of SirT1 to support the effect of caloric restriction in aging muscle highlight that it is not necessary for young muscle. Moreover, regardless of SirT1 function, the paper indicates integrin signaling, EGF receptor signaling, Wnt signaling, and nicotinic acetylcholine receptor signaling as pathways regulated by caloric restriction.
The study has been nicely performed and designed even if the paper is full of tables that make the reading discontinuous. I wondered if some tables/figures could be better grouped or moved to supplementary materials.
I suggest to reorganize what written above.
Author Response
The paper "Skeletal Muscle Gene Expression Profile In Response to Ca-2 loric Restriction During Aging: A role for SirT1" by Myers et al. suggests the role of SirT1 to support the effect of caloric restriction in aging muscle highlight that it is not necessary for young muscle. Moreover, regardless of SirT1 function, the paper indicates integrin signaling, EGF receptor signaling, Wnt signaling, and nicotinic acetylcholine receptor signaling as pathways regulated by caloric restriction. The study has been nicely performed and designed even if the paper is full of tables that make the reading discontinuous. I wondered if some tables/figures could be better grouped or moved to supplementary materials. I suggest to reorganize what written above.
Since we have reanalyzed the data by removing bad quality and odd reads, we hope the revised manuscript would be interesting to the reviewers and readers. We have included tables that only have significantly expressed gene data.
Reviewer 3 Report
Please find enclosed my comments concerning the manuscript entitled “Skeletal Muscle Gene Expression Profile In Response to Caloric Restriction During Aging: A role for SirT1”
This study investigated the role of Calorie restriction on the genome-wide gene expression profile in the skeletal muscle from young versus old mice in the presence or absence of Sirtuin-1.
The subject is very interesting, however, I have some comments aimed to improve the information given in the paper before being published.This manuscript has several important weaknesses that should be corrected.
ABSTRACT
The abstract must be complete. Several information are missing: the % of calorie restriction (CR), the duration of CR.
INTRODUCTION
The introduction is well written. However, which are the main functions in muscle which are regulated by sirtuin1.
METHODS
Please include a diagram of our different group and treatments. I think a CR of 60% is very important for mice. How did you measure the calorie intake? Your study includes males and/or females mice?, and justify.
RESULTS
There are many interesting results but it is difficult to understand them and to connect them to each other
Could you group the different genes according to the functions regulated by sirtuin-1 at the muscle level according to the experimental groups
DISCUSSION
The discussion is too short
You mentioned that several genes are significantly altered by CR but it is not clear what are the physiological consequences in mice. A synthesis diagram regrouping the different functions regulated by sirt1 at the muscle level according to age should be included
Author Response
ABSTRACT
The abstract must be complete. Several information are missing: the % of calorie restriction (CR), the duration of CR.
We have included these pieces of information in the revised manuscript.
INTRODUCTION
The introduction is well written. However, which are the main functions in muscle which are regulated by sirtuin1.
Our revised results and discussion sections have been mainly focused to explore the role of SirT1 in the regulation of CR responsive genes associated with skeletal muscle homeostasis.
METHODS
Please include a diagram of our different group and treatments. Table 1 has this information. I think a CR of 60% is very important for mice. How did you measure the calorie intake? Your study includes males and/or females mice?, and justify.
We measured the CR based on the total amount of food intake by measuring the remaining leftover diet each day for two weeks. We have used an equal number of both male and female mice. Mentioned in the revised manuscript.
RESULTS
There are many interesting results but it is difficult to understand them and to connect them to each other.
Could you group the different genes according to the functions regulated by sirtuin-1 at the muscle level according to the experimental groups.
In the revised manuscript, our new data showed only SirT1 and/or CR responsive pathways associated with skeletal muscle function.
DISCUSSION
The discussion is too short.
We have stretched and improved the discussion significantly.
You mentioned that several genes are significantly altered by CR but it is not clear what are the physiological consequences in mice. A synthesis diagram regrouping the different functions regulated by sirt1 at the muscle level according to age should be included.
New supplemental tables S3-S13 have the data specific to SirT1, CR and age. Most of the pathways we used in these tables have been focused on skeletal muscle function.
Round 2
Reviewer 1 Report
The authors have addressed all the raised concerns during the first round of revision and have done tremendous improvements in the manuscript. The new analysis performed in the revised manuscript provides a simplified view of Sirt1-regulated transcriptomic changes in skeletal muscles in aging mice upon caloric restriction. The manuscript will be perceived well by the readers.
Reviewer 2 Report
I appreciate the reorganization of the paper which has improved in quality. Dividing the results into subsections makes them more usable even if, in my opinion, too many tables continue to be in the main text. The discussion is also implemented and enriched. However some problems are still present: for instance line 510-11 seems an incomplete sentence , some repetitions and some typos.
Reviewer 3 Report
For me, this manuscript is now acceptable for publication. The authors have structured it much better in order to highlight the explorations conducted.